# GemNet-OC: Developing Graph Neural Networks for Large and Diverse Molecular Simulation Datasets

**Johannes Gasteiger**                                    *johannes.gasteiger@tum.de*
*Technical University of Munich*
*Work partially done during an internship at FAIR, Meta AI.*

**Muhammed Shuaibi**                                      *mshuaibi@fb.com*
*Carnegie Mellon University → FAIR, Meta AI*

**Anuroop Sriram**                                       *anuroops@fb.com*
*FAIR, Meta AI*

**Stephan Günnemann**                                    *stephan.guennemann@tum.de*
*Technical University of Munich*

**Zachary Ulissi**                                        *zulissi@andrew.cmu.edu*
*Carnegie Mellon University*

**C. Lawrence Zitnick**                                   *zitnick@fb.com*
*FAIR, Meta AI*

**Abhishek Das**                                          *abhshkdz@fb.com*
*FAIR, Meta AI*

**Reviewed on OpenReview:** *https://openreview.net/forum?id=u8tvSxm4Bs*

## Abstract

Recent years have seen the advent of molecular simulation datasets that are orders of magnitude larger and more diverse. These new datasets differ substantially in four aspects of complexity: 1. Chemical diversity (number of different elements), 2. system size (number of atoms per sample), 3. dataset size (number of data samples), and 4. domain shift (similarity of the training and test set). Despite these large differences, benchmarks on small and narrow datasets remain the predominant method of demonstrating progress in graph neural networks (GNNs) for molecular simulation, likely due to cheaper training compute requirements. This raises the question – *does GNN progress on small and narrow datasets translate to these more complex datasets?* This work investigates this question by first developing the GemNet-OC model based on the large Open Catalyst 2020 (OC20) dataset Chanussot et al. (2021). GemNet-OC outperforms the previous state-of-the-art on OC20 by 16 % while reducing training time by a factor of 10. We then compare the impact of 18 model components and hyperparameter choices on performance in multiple datasets. We find that the resulting model would be drastically different depending on the dataset used for making model choices. To isolate the source of this discrepancy we study six subsets of the OC20 dataset that individually test each of the above-mentioned four dataset aspects. We find that results on the OC-2M subset correlate well with the full OC20 dataset while being substantially cheaper to train on. Our findings challenge the common practice of developing GNNs solely on small datasets, but highlight ways of achieving fast development cycles and generalizable results via moderately-sized, representative datasets such as OC-2M and efficient models such as GemNet-OC. Our code and pretrained model weights are open-sourced at `github.com/Open-Catalyst-Project/ocp`.

# 1    Introduction

Machine learning models for molecular systems have recently experienced a leap in accuracy with the advent of graph neural networks (GNNs) (Gilmer et al., 2017; Schütt et al., 2017; Gasteiger et al., 2020b; Batzner et al., 2021; Ying et al., 2021). However, this progress has predominantly been demonstrated on datasets covering a limited set of systems (Ramakrishnan et al., 2014) – sometimes even just single molecules (Chmiela et al., 2017). To scale this success to large and diverse atomic datasets and real-world chemical experiments, models need to demonstrate their abilities along four orthogonal aspects: chemical diversity, system size, dataset size, and test set difficulty. The critical question then becomes: *Do model improvements demonstrated on small and limited datasets generalize to large and diverse molecular systems?*

Some results suggest that they should indeed generalize. GNN architectures for molecular simulations usually generalize between systems and datasets (Unke & Meuwly, 2019). Moreover, models have been found to scale well with training set size (Bornschein et al., 2020). However, other works found that model changes can indeed break this behavior, albeit positing that this is limited to singular model properties (Batzner et al., 2021). It is generally accepted that certain hyperparameters should be adapted to each dataset, e.g. the learning rate, batch size, embedding size. However, there is no qualitative reason why only these "special" properties would be affected by the underlying dataset, while other aspects of model components would be unaffected. Changes in model trends between datasets have already been seen in other fields, e.g. in computer vision (Kornblith et al., 2019).

In this work, we set out to directly test the performance of model components and hyperparameters between datasets. We focus on changes around one baseline model instead of separate GNN architectures, since this most closely matches the fine-grained changes that are typical for model development. We develop the GemNet-OC baseline model based on the Open Catalyst 2020 dataset (OC20) (Chanussot et al., 2021), which is the largest molecular dataset to date. The resulting GemNet-OC model is 10 times cheaper to train than previous models and outperforms the previous state-of-the-art by 16 %. It incorporates optimizations that enable the calculation of quadruplet interactions and dihedral angles even in large systems, and introduces an interaction hierarchy to better model long-range interactions.

We analyze these new model components, as well as previously proposed components and hyperparameters, on two narrow datasets (MD17 (Chmiela et al., 2017) and COLL (Gasteiger et al., 2020a)), the large-scale OC20 dataset, and six OC20 subsets that isolate the effects of the aforementioned four dataset properties. We find that model components can indeed have substantially different effects on different datasets, and that all four dataset properties can cause such differences. Large and diverse datasets like OC20 thus pose a learning challenge that is *qualitatively different* from the smaller chemical spaces and structures represented in most prior molecular datasets.

Testing a model that can generalize well to the large diversity of chemistry thus requires a sufficiently large and diverse dataset. However, model development on large datasets like OC20 is excessively expensive due to long training times. As a case in point, the analyses and models presented in this work required more than 16 000 GPU days of training overall. To solve this issue, we identify a small data subset on which model trends correlate well with OC20: OC-2M. We provide multiple baseline results for OC-2M to enable its use in future work. Together, GemNet-OC and OC-2M enable state-of-the-art model development even on a single GPU. Our code and pretrained model weights are open-sourced at `github.com/Open-Catalyst-Project/ocp`. Concretely, our contributions are:

- We propose GemNet-OC, a GNN that achieves state-of-the-art results on all OC20 tasks while training is 10 times faster than previous large models.
- Through carefully-controlled experiments on a variety of datasets, we demonstrate the discrepancy between modeling decisions on small and limited vs. large and diverse datasets.
- We identify OC-2M, a small data subset that provides generalizable model insights, and publish a range of baseline results to enable its use in future work.

## 2    Background and related work

**Learning atomic interactions.** A model for molecular simulations takes $N$ atoms, their atomic numbers $\boldsymbol{z} = \{z_1, \ldots, z_N\}$, and positions $\boldsymbol{X} \in \mathbb{R}^{N \times 3}$ and predicts the system's energy $E$ and the forces $\boldsymbol{F} \in \mathbb{R}^{N \times 3}$ acting on each atom. Note that these models usually do not use further auxiliary input information such as bond types, since these might be ill-defined in certain states. The force predictions are then used in a simulation to calculate the atoms' accelerations during a time step. Alternatively, we can find low-energy or relaxed states of the system by performing geometric optimization based on the atomic forces. More concretely, the atomic forces $\boldsymbol{F}$ are the energy's gradient, i.e. $\boldsymbol{F}_i = -\frac{\partial}{\partial \boldsymbol{x}_i} E$. The forces are thus conservative, which is important for the stability of long molecular dynamics simulations since it ensures energy conservation and path independence. However, in some cases we can ignore this requirement and predict forces directly, which considerably speeds up the model (Park et al., 2021). Classical approaches for molecular machine learning models use hand-crafted representations of the atomic neighborhood (Bartók et al., 2013) with Gaussian processes (Bartók et al., 2010; 2017; Chmiela et al., 2017) or neural networks (Behler & Parrinello, 2007; Smith et al., 2017). However, these approaches have recently been surpassed consistently by graph neural networks (GNNs), in both low-data and large-data regimes (Batzner et al., 2021; Gasteiger et al., 2021).

**Graph neural networks.** GNNs represent atomic systems as graphs $\mathcal{G} = (\mathcal{V}, \mathcal{E})$, with atoms defining the node set $\mathcal{V}$. The edge set $\mathcal{E}$ is typically defined as all atoms pairs within a certain cutoff distance, e.g. 5 Å. The first models resembling modern GNNs were proposed by Sperduti & Starita (1997); Baskin et al. (1997). However, they only became popular after multiple works demonstrated their potential for a wide range of graph-related tasks (Bruna et al., 2014; Kipf & Welling, 2017; Gasteiger et al., 2019). Molecules have always been a major application for GNNs (Baskin et al., 1997; Duvenaud et al., 2015), and molecular simulation is no exception. Modern examples include SchNet (Schütt et al., 2017), PhysNet (Unke & Meuwly, 2019), Cormorant (Anderson et al., 2019), DimeNet (Gasteiger et al., 2020b), PaiNN (Schütt et al., 2021), SpookyNet (Unke et al., 2021), and SpinConv (Shuaibi et al., 2021). Notably, MXMNet (Zhang et al., 2020) proposes a two-level message passing scheme. Similarly, Alon & Yahav (2021) propose a global aggregation mechanism to fix a bottleneck caused by GNN aggregation. GemNet-OC extends these two-level schemes to a multi-level interaction hierarchy, and leverages both edge and atom embeddings.

**Model trends between datasets.** Previous work often found that models scale equally well with training set size (Hestness et al., 2017). This lead to the hypothesis that model choices on subsets of a dataset translate well to the full dataset (Bornschein et al., 2020). In contrast, we find that model choices can have *different* effects on small and large datasets, which implies *different scaling* for different model variants. Similarly, Brigato & Iocchi (2020) found that simple models work better than state-of-the-art architectures in the extremely small dataset regime, and Kornblith et al. (2019) observed differences in model trends between datasets for computer vision. Note that we observe differences for subsets of the *same* dataset, and our findings do not require a dataset reduction to a few dozen samples — we observe qualitative differences even beyond 200 000 samples.

## 3    Datasets

**Datasets for molecular simulation.** Training a model for molecular simulation requires a dataset with the positions and atomic numbers of all atoms for the model input and the forces and energy as targets for its output. The energy is either the total inner energy, i.e. the energy needed to separate all electrons and nuclei, or the atomization energy, i.e. the energy needed to separate all atoms. Note that the energy is actually not necessary for learning a consistent force field. Most works weight the forces much higher than the energy or even train only on the forces.

Many molecular datasets have been proposed in recent years, often with the goal of supporting model development for specific use cases. Arguably the most prominent, publicly available datasets are MD17 (Chmiela et al., 2017), ISO17 (Schütt et al., 2017), $S_N2$ (Unke & Meuwly, 2019), ANI-1x (Smith et al., 2020), QM7-X (Hoja et al., 2020), COLL (Gasteiger et al., 2020a), and OC20 (Chanussot et al., 2021). Table 1 gives an overview of these datasets. Note that datasets such as QM9 (Ramakrishnan et al., 2014) or OQMD (Saal et al., 2013) cannot be used for learning molecular simulations since they only contain systems at equilibrium, where all forces are zero. Other datasets such as DES370k (Donchev et al., 2021) and OrbNet Denali (Christensen et al., 2021)

Table 1: Common molecular simulation datasets. Datasets prior to OC20 only cover (1) a narrow range of elements (and molecules), and consequently a low number of neighbor pairs (defined as distinct element pairs within 5 Å), and (2) small systems. Furthermore, they (3) provide far fewer samples and (4) often use a test set that is correlated with the training set since they consist of data from the same simulation trajectories. We investigate six OC20 subsets to isolate these effects.

| Dataset | Description | Elements | Neighbor pairs | Avg. size | Train set size | Test set(s) |
|---|---|---|---|---|---|---|
| MD17 | Eight separate molecules | H, C, N, O | 3-10 | 12.5 (9-21) | 8 × 1000 | Same, single trajectory |
| ISO17 | $C_7O_2H_{10}$ isomers | H, C, O | 6 | 19 | 404 000 | Same traj. / OOD systems |
| $S_N2$ | Methyl halides, halide ions | H, C, F, Cl, Br, I | 20 | 5.4 (2-6) | 400 000 | Same trajectories |
| ANI-1x | Selected MD samples | H, C, N, O | 10 | 15.3 (2-63) | 4 956 005 | OOD systems (COMP6) |
| QM7-X | Small molecules | H, C, N, O, S, Cl | 20 | 16.7 (4-23) | 4 175 037 | Same traj. / OOD systems |
| COLL | Molecule collisions | H, C, O | 6 | 10.2 (2-26) | 120 000 | Same trajectories |
| OC20 | Relaxations of catalysts | 56 | 1454 | 73.3 (7-225) | 133 934 018 | Sep. traj. / OOD ads.+cat. |
| OC-Rb | Only H, C, N, O, Rb | H, C, N, O, Rb | 15 | 39.1 (7-220) | 524 736 | Sep. traj. / OOD adsorbates |
| OC-Sn | Only H, C, N, O, Sn | H, C, N, O, Sn | 15 | 59.5 (22-220) | 257 757 | Sep. traj. / OOD adsorbates |
| OC-sub30 | At most 30 atoms | 55 | 881 | 24.6 (7-30) | 4 020 568 | Sep. traj. / OOD ads.+cat. |
| OC-200k | Random subset | 56 | 1454 | 73.2 (7-225) | 200 000 | Sep. traj. / OOD ads.+cat. |
| OC-2M | Random subset | 56 | 1454 | 73.3 (7-225) | 2 000 000 | Sep. traj. / OOD ads.+cat. |
| OC-XS | ≤ 30 H, C, N, O, Rb atoms | H, C, N, O, Rb | 15 | 19.7 (7-30) | 298 797 | Same / sep. traj. / OOD ads. |

contain out-of-equilibrium systems but do not provide force labels. This makes them ill-suited for this task as well, since energies only provide one label per sample while forces provide $3N$ labels (Christensen & Lilienfeld, 2020). In this work we focus on the OC20 dataset, which consists of single adsorbates (small molecules) physically binding to the surfaces of catalysts. The simulated cell of $N$ atoms uses periodic boundary conditions to emulate the behavior of a crystal surface.

**Dataset aspects.** The difficulty and complexity of molecular datasets can largely be divided into four aspects: chemical diversity, system size, dataset size, and domain shift. Chemical diversity (number of atom types or interacting pairs) and system size (number of independent atoms) determine the data's complexity and thus the expressive power a model needs to fit this data. The dataset size determines the amount of data from which the model can learn. Note that a large dataset might still need a very sample-efficient model if the underlying data is very complex and diverse. Finally, we look at the test set's domain shift to determine its difficulty. The learning task might be significantly easier than expected if the test set is very similar to the training set. Note that many dataset aspects are not covered by these four properties, such as the method used for calculating the molecular data or the types of systems. OC20 has multiple details that fall into this category: It consists of periodic systems of a bulk material with an adsorbate on its surface, and has a canonical z-axis and orientation. These details also affect the importance of modeling choices, such as rotation invariance (Hu et al., 2021). We chose to focus on the above four properties since they are easy to quantify, apply to every dataset, and have a large effect by themselves.

**Data complexity.** Chemical diversity and system size determine the complexity of the underlying data: How many different atomic environments does the model need to fit? Are there long-range interactions or collective many-body effects caused by large systems? To quantify a dataset's chemical diversity we count the number of elements and the element pairs that occur within a range of 5 Å ("neighbor pairs"). However, note that these proxies are not perfect. For example, COLL only contains three elements but samples a significantly larger region of conformational space than QM7-X. Subsequently, SchNet only achieves a force MAE of 172 eV/Å on COLL, but 53.7 eV/Å on QM7-X (same trajectory). Still, these proxies do illustrate the stark difference between OC20 and other datasets: OC20 has 70× more neighbor pairs than other datasets (Table 1).

**Dataset size.** Dataset size determines how much information is available for a model to learn from. Note that larger systems also increase the effective dataset size since each atom provides one force label. Usually, the dataset size should just be appropriately chosen to reach good performance for a given dataset complexity. Table 1 thus lists the official or a typical training set size for each dataset, and not the total dataset size. An extreme outlier in this respect is MD17. In principle it has 3 611 115 samples, i.e. 450 000 samples per each of the 8 molecule trajectories on average. This is extremely large for the simple task of fitting single small

molecules close to their equilibrium. Most recent works thus only train on 1000 MD17 samples per molecule, i.e. less than 1 % of the dataset. Even in this setting, modern models fit the DFT data more closely than the DFT method fits the ground truth (Gasteiger et al., 2020b).

**Domain shift.** A dataset's test set is another important aspect that determines a dataset's difficulty and how well it maps to real-world problems. In practice we apply a trained model to a new simulation, i.e. a different simulation trajectory than it was trained on. We might even want to apply it to a different, out-of-distribution (OOD) system. For example, the OC20 dataset contains OOD test sets with unseen adsorbates (ads.), catalysts (cat.) and both (ads.+cat.). However, many datasets are created by running a set of simulations and then randomly splitting up the data into the training and test sets. The data is thus taken from the *same trajectories*, which leads to strongly correlated training and test data: the test samples will never be significantly far away from the training samples. To properly decorrelate test samples, we have to either let enough simulation time pass between the training and test samples or use a separate simulation trajectory (sep. traj.). An extreme case in this aspect is again MD17: its test samples are taken from the same, single trajectory and the same, single molecule as the training set. MD17 thus only measures minor generalization across conformational space, not chemical space. This severely limits the significance of results on MD17. Moreover, improvements demonstrated on MD17 might not even be useful due to the data's limited accuracy. The most likely reason for MD17's pervasiveness is that its small size makes model development fast and efficient, coupled with the hypothesis that discovered model improvements would work equally well on more complex datasets. Unfortunately, we found this hypothesis to be incorrect, as we show in more detail below. This raises the question: *Which dataset aspect changes model performance between benchmarks?* Answering this would allow us to create a dataset that combines the best of both worlds: fast development cycles and results that generalize to realistic challenges like OC20.

**OC20 subsets.** We investigate six subsets of the OC20 dataset to isolate the effects of each dataset aspect. Comparing subsets of the same dataset ensures that the observed differences are only due to these specific aspects, and not due to other dataset properties. OC-Rb and OC-Sn reduce chemical diversity by restricting the dataset to a limited set of catalyst materials. We chose rubidium and tin since they are among the most frequent catalyst elements in OC20. We would expect similar results for other catalyst elements. OC-sub30 isolates the effect of small systems by only allowing up to 30 atoms per system. Note that this does not influence the number of neighbors per atom, since OC20 uses periodic boundary conditions. It does, however, impact its effective training size and might skew the selection of catalysts. OC-200k and OC-2M use random subsets of OC20, thus isolating the effect of dataset size. Finally, OC-XS investigates the combined effect of restricting the system size to 30 independent atoms and only using Rubidium catalysts. This also naturally decreases the dataset size. We apply these subsets to both the OC20 training and validation set in the same way. To investigate the effect of test set choice we additionally investigate OOD validation splits by taking the respective subsets of the OC20 OOD dataset as well. We use OOD catalysts and adsorbates ("OOD both") for OC-sub30, OC-200k, and OC-2M, and only OOD adsorbates for OC-Rb, OC-Sn, and OC-XS, since no OOD catalysts are available for these subsets. Additionally, we introduce a third validation set for OC-XS by splitting out random samples of the training trajectories. This selection mimicks the easier "same trajectory" test sets used e.g. by MD17 (Chmiela et al., 2017) and COLL (Gasteiger et al., 2020a).

## 4 GemNet-OC

We investigate the effect of datasets on model choices by first developing a GNN on OC20, starting from the geometric message passing neural network (GemNet) (Gasteiger et al., 2021). While regular message passing neural networks (MPNNs) only embed each atom $a$ as $\boldsymbol{h}_a \in \mathbb{R}^H$ (Gilmer et al., 2017), GemNet additionally embeds the directed edges between atoms as $\boldsymbol{m}_{(ba)} \in \mathbb{R}^{H_\mathrm{m}}$. Both embeddings are then updated in multiple learnable layers using neighboring atom and edge embeddings and the full geometric information – the distances between atoms $x_{ba}$, the angles between neighboring edges $\varphi_{cab}$, and the dihedral angles defined via triplets of edges $\theta_{cabd}$. We call our new model GemNet-OC. GemNet-OC is well representative of previous models. It uses a similar architecture and the atomwise force errors are well correlated: The Spearman rank correlation to SchNet's force errors is 0.44, to DimeNet$^{++}$ 0.50, to SpinConv 0.46, and to GemNet 0.59. It is just slightly higher for a separately trained GemNet-OC model, at 0.66. We next describe the components we propose and investigate in GemNet-OC.

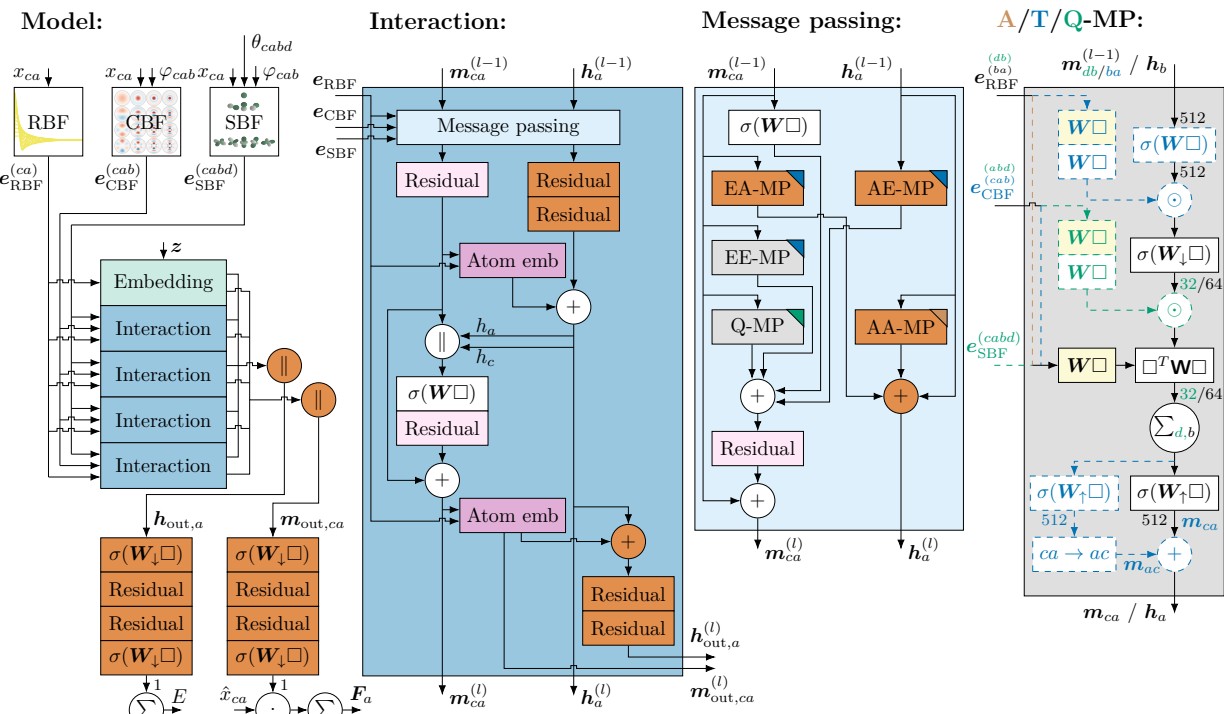

Figure 1: Main parts of the GemNet-OC architecture. Changes are highlighted in orange. □ denotes the layer's input, ∥ concatenation, $\sigma$ a non-linearity, and yellow a layer with weights shared across interaction blocks. Numbers denote embedding sizes. Whether the input or output of the MP-block are atom or edge embeddings depends on the type of message passing. Difference between different variants (AA, AE, EA, AA, Q-MP) are shown via colors and dashed lines. Note that triplet components are also included in quadruplet interactions.

**Neighbors instead of cutoffs.** GNNs for molecules typically construct an interatomic graph by connecting all atoms within a cutoff of 5 Å to 6 Å. Distances are then usually represented using a radial basis, which is multiplied with an envelope function to ensure a twice continuously differentiable function at the cutoff (Gasteiger et al., 2020b). This is required for well-defined force training if the forces are predicted via backpropagation. However, when we face the large chemical diversity of a dataset like OC20 there are systems where atoms are typically farther apart than 6 Å, and other systems where all neighboring atoms are closer than 3 Å. Using one fixed cutoff can thus cause a disconnected graph in some cases, which is detrimental for energy predictions, and an excessively dense graph in others, which is computationally expensive. To solve this dilemma, we propose to construct a graph from a fixed number of nearest neighbors instead of using a distance cutoff. This might initially seem problematic since it breaks differentiability for forces $F = -\frac{\partial}{\partial \boldsymbol{x}} E$ if two atoms switch order by moving some small $\varepsilon$. However, we found that this is not an issue in practice. The nearest neighbor graph leads to the same or better accuracy, *triples* GemNet-OC's throughput, provides easier control of computational and memory requirements, and ensures a consistent, fixed neighborhood size.

**Simplified basis functions.** As proposed by Gasteiger et al. (2020b), GemNet represents distances $x_{ba}$ using spherical Bessel functions $j_l(\frac{z_{ln}}{c_{int}} x_{ba})$ of order $l$, with the interaction cutoff $c_{int}$ and the Bessel function's $n$-th root $z_{ln}$, and angular information using real spherical harmonics $Y_m^{(l)}(\varphi_{cab}, \theta_{cabd})$ of degree $l$ and order $m$. Note that the radial basis order $l$ is *coupled* to the angular basis degree $l$. This basis thus requires calculating $nl$ radial and $lm$ angular functions. This becomes expensive for large systems with a large number of neighbors. If we embed $k_{emb}$ edges per atom and compute dihedral angles for $k_{qint}$ neighbors, we need to calculate $\mathcal{O}(Nk_{qint}k_{emb}^2)$ basis functions. To reduce the basis's computational cost, we first decouple the radial basis from the angular basis by using Gaussian or 0-order Bessel functions, independent of the spherical harmonics. We then streamline the spherical harmonics by instead using an outer product of order 0 spherical harmonics, which simplify to Legendre polynomials $Y_0^{(l)}(\varphi_{cab})Y_0^{(m)}(\theta_{cabd}) = P_l(\cos \varphi_{cab})P_m(\cos \theta_{cabd})$. This

only requires the normalized inner product of edge directions, not the angle. These simplified basis functions increase throughput by 29 %, without hurting accuracy.

**Tractable quadruplet interactions.** Next, we tackle GemNet's dihedral angle-based interactions for large systems. These 'quadruplet interactions' are notoriously expensive, since they scale as $\mathcal{O}(Nk_{\text{qint}}k_{\text{emb}}^2)$. However, we observed that quadruplet interactions are mostly relevant for an atom's closest neighbors. Their benefits quickly flatten out as we increase $k_{\text{qint}}$. We thus choose a low $k_{\text{qint}} = 8$. This is substantially lower than our $k_{\text{emb}} = 30$, since we found model performance to be rather sensitive to $k_{\text{emb}}$. This is opposite to the original GemNet, where the quadruplet cutoff was larger than the embedding cutoff. Quadruplet interactions only cause an overhead of 31 % thanks to these optimizations, instead of the original 330 % (Gasteiger et al., 2021).

**Interaction hierarchy.** Using a low number of quadruplet interactions essentially introduces a hierarchy of expensive short-distance quadruplet interactions and cheaper medium-distance edge-to-edge interactions. We propose to extend this interaction hierarchy further by passing messages between the atom embeddings $\boldsymbol{h}_a$ directly. This update has the same structure as GemNet's message passing, but only uses the distance between atoms. This atom-to-atom interaction is very cheap due to its complexity of $\mathcal{O}(Nk_{\text{emb}})$. We can thus use a long cutoff of 12 Å without any neighbor restrictions. To complement this interaction, we also introduce atom-to-edge and edge-to-atom message passing. These interactions also follow the structure of GemNet's original message passing. Each adds an overhead of roughly 10 %.

**Further architectural improvements.** We make three architectural changes to further improve GemNet-OC. First, we output an embedding instead of directly generating a separate prediction per interaction block. These embeddings are then concatenated and transformed using multiple learnable layers (denoted as MLP) to generate the overall prediction. This allows the model to better leverage and combine the different pieces of information after each interaction block. Second, we improve atom embeddings by adding a learnable MLP in each interaction block. This block is beneficial due to the new atom-to-atom, edge-to-atom, and atom-to-edge interactions, which directly use the atom embeddings. Third, we improve the usage of atom embeddings further by adding them to the embedding in each energy output block. This creates a direct path from atom embeddings to the energy prediction, which improves the information flow. These model changes add less than 2 % computational overhead. See Fig. 1 for an overview of the GemNet-OC model.

## 5 Results on OC20

**OC20 tasks.** In the OC20 benchmark, predicting energies and forces is the first of three tasks, denoted as structure to energy and forces (S2EF). There are two important additional scores for this task besides the energy and force MAEs. The force cosine determines how correct the predicted force *directions* are and energy and forces within threshold (EFwT) measures how many structures are predicted "correctly", i.e. have predicted energies and forces sufficiently close to the ground-truth. The second task is initial structure to relaxed structure (IS2RS). In IS2RS we perform energy optimization from an initial structure using our model's force predictions as gradients. After the optimization process we measure whether the distance between our final relaxed structure and the true relaxed structure is below a variety of thresholds (average distance within threshold, ADwT), and whether additionally the structure's true forces are close to zero (average force below threshold, AFbT). The third task, initial structure to relaxed energy (IS2RE), is to predict the energy of this relaxed structure. An alternative to this relaxation-based approach is the so-called direct IS2RE approach. Direct models learn and predict energies directly using only the initial structures. This approach is much cheaper, but also less accurate. Our work focuses on the regular relaxation-based approach, which is centered around S2EF data and models.

**Results.** In Table 2 we provide test results for all three OC20 tasks, as well as S2EF validation results, averaged over the in-distribution (ID), OOD-adsorbate, OOD-catalyst, and OOD-both datasets. To facilitate and accelerate future research, we provide multiple new baseline results on the smaller OC-2M subset for SchNet (Schütt et al., 2017), DimeNet$^{++}$ (Gasteiger et al., 2020a), SpinConv (Shuaibi et al., 2021), and GemNet-dT (Gasteiger et al., 2021). OC-2M is smaller but still well-representative of model choices on full OC20, as we will discuss in Sec. 6. We use the same OC20 test set for both the OC-2M and OC20 training sets. On full OC20 we additionally compare to CGCNN (Xie & Grossman, 2018), ForceNet (Hu et al., 2021),

Table 2: Training throughput and results for the validation set and the three test tasks of OC20, averaged across all four splits. GemNet-OC-L outperforms prior models by 16 %. GemNet-OC trained on OC-2M outperforms all pre-GemNet models trained on the full OC20 dataset (∼134 M samples).

| Train set | Model | Throughput Samples / GPU sec. ↑ | S2EF validation Energy MAE meV ↓ | Force MAE meV/Å ↓ | Force cos ↑ | EFwT % ↑ | S2EF test Energy MAE meV ↓ | Force MAE meV/Å ↓ | Force cos ↑ | EFwT % ↑ | IS2RS AFbT % ↑ | ADwT % ↑ | IS2RE Energy MAE meV ↓ |
|---|---|---|---|---|---|---|---|---|---|---|---|---|---|
| OC-2M | SchNet | - | 1400 | 78.3 | 0.109 | 0.00 | 1370 | 77.1 | 0.116 | 0.00 | - | - | - |
| | DimeNet++ | - | 805 | 65.7 | 0.217 | 0.01 | 761 | 63.0 | 0.222 | 0.01 | - | - | - |
| | SpinConv | - | 406 | 36.2 | 0.479 | 0.13 | 401 | 35.5 | 0.475 | 0.13 | - | - | - |
| | GemNet-dT | - | 358 | 29.5 | 0.557 | 0.61 | 323 | 28.1 | 0.559 | 0.69 | 16.7 | 54.8 | 438 |
| | GemNet-OC | - | **286** | **25.7** | **0.598** | **1.06** | **274** | **24.3** | **0.603** | **1.25** | **19.6** | **56.4** | **407** |
| OC20 | CGCNN | - | 590 | 74.0 | 0.142 | 0.01 | 608 | 73.3 | 0.146 | 0.01 | - | - | - |
| | SchNet | - | 549 | 56.8 | 0.297 | 0.06 | 540 | 54.7 | 0.302 | 0.06 | - | 14.4 | 764 |
| | ForceNet-large | 15.3 | - | 33.5 | 0.515 | - | - | 32.0 | 0.516 | 0.01 | 12.7 | 49.6 | - |
| | DimeNet++-L-F+E | 4.6 | 515 | 32.8 | 0.541 | 0.00 | 480 | 31.3 | 0.544 | 0.00 | 21.7 | 51.7 | 559 |
| | PaiNN | 60.0 | - | - | - | - | 341 | 33.1 | 0.491 | 0.46 | 11.7 | 48.5 | 471 |
| | SpinConv | 6.0 | 371 | 41.2 | 0.473 | 0.05 | 336 | 29.7 | 0.539 | 0.45 | 16.7 | 53.6 | 437 |
| | GemNet-dT | 25.8 | 315 | 27.2 | 0.594 | 0.54 | 292 | 24.2 | 0.616 | 1.20 | 27.6 | 58.7 | 400 |
| | GemNet-XL | 1.5 | - | - | - | - | 270 | **20.5** | 0.660 | 1.81 | 30.8 | **62.7** | 371 |
| | GemNet-OC | 18.3 | **244** | **21.7** | **0.662** | **2.07** | **233** | 20.7 | **0.666** | **2.50** | **35.3** | 60.3 | **355** |
| OC20+ OC-MD | GemNet-OC-L-E | 7.5 | **239** | 22.1 | 0.662 | 2.30 | **230** | 21.0 | 0.665 | 2.80 | - | - | - |
| | GemNet-OC-L-F | 3.2 | 252 | **20.0** | **0.687** | **2.51** | 241 | **19.0** | **0.691** | **2.97** | 40.6 | 60.4 | - |
| | GemNet-OC-L-F+E | - | - | - | - | - | - | - | - | - | - | - | **348** |

PaiNN (Schütt et al., 2021), and GemNet-XL (Sriram et al., 2022). Note that PaiNN uses our independent reimplementation of the original PaiNN architecture with the difference that forces are predicted directly from vectorial features via a gated equivariant block instead of gradients of the energy output. This breaks energy conservation but is essential for good performance on OC20. GemNet-Q (Gasteiger et al., 2021) runs out of memory when training on OC20. GemNet-OC outperforms all previous models on both the OC-2M and OC20 training sets. GemNet-OC on OC-2M even performs on par with GemNet-dT on OC20, while using *70 times less training data*. It also performs better than all previously proposed direct IS2RE models, such as 3D-Graphormer (Ying et al., 2021), which achieves an IS2RE MAE of 472.2 meV. Direct models have seen a lot of interest due to their fast training and development. The OC-2M S2EF dataset provides a similarly fast development method for relaxation-based models. After model selection, we can scale the model up to larger training sets and model sizes. We demonstrate this with the GemNet-OC-L model, which was trained on both the full OC20 and the OC-MD datasets. OC-MD complements the regular OC20 dataset with data points from molecular dynamics. We trained one model focussing on energy predictions (GemNet-OC-L-E) and a second one for force predictions (GemNet-OC-L-F). For IS2RE we then combined both in the GemNet-OC-L-F+E model. These models set the state of the art for *all* OC20 tasks. GemNet-OC even outperforms GemNet-XL, despite GemNet-XL being substantially larger and slower to train.

**Training time.** Fig. 2 shows that GemNet-OC surpasses the accuracy of GemNet-dT after 600 GPU hours. It is 40 % slower per sample than a performance-optimized version of GemNet-dT. However, if we consider that GemNet-OC uses four interaction blocks instead of three as in GemNet-dT, we see that the GemNet-OC architecture is roughly as fast as GemNet-dT overall. GemNet-OC-L is roughly 4.5 times slower per sample but still converges faster, surpassing regular GemNet-OC after 2800 GPU hours. GemNet-OC-L reaches the final accuracy of GemNet-XL in 10 times fewer GPU hours, and continues to improve further. Combined with GemNet-OC's good results on the OC-2M dataset, this fast convergence allows for much faster model development. Overall, GemNet-OC shows that we can substantially improve a model like GemNet if we focus development on the

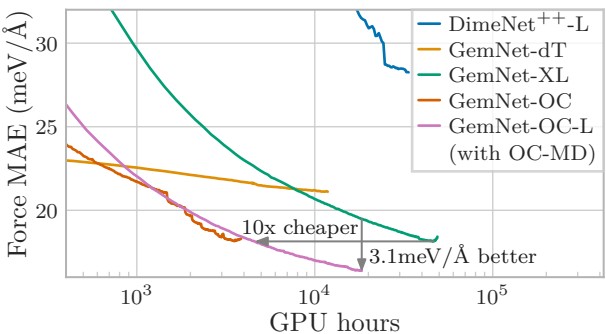

Figure 2: Convergence of GemNet-OC and previous models. GemNet-OC surpasses GemNet-dT after 600 GPU hours, and is ∼12x faster than GemNet-XL.

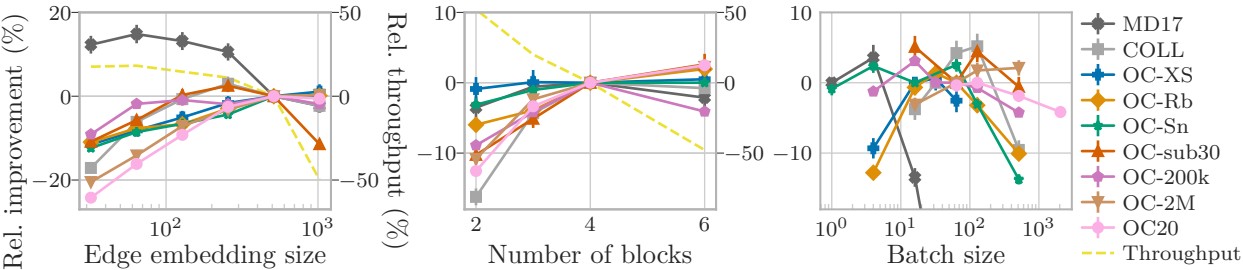

Figure 3: Effect of batch size, edge embedding size, and number of blocks on force MAE and throughput. Most datasets are insensitive to the batch size above a certain size. Large datasets strongly benefit from large models, while small datasets exhibit optima at small sizes.

target dataset. However, this does not yet answer the question of how its model choices would change if we focused on another dataset. We will investigate this question in the next section.

## 6 Model trends across datasets

For the purpose of comparing model choices between datasets we run all ablations and senstivity analyses on the Aspirin molecule of MD17, the COLL dataset, the full OC20 dataset, and the OC-Rb, OC-Sn, OC-sub30, OC-200k, OC-2M, and OC-XS data subsets. For consistency we use the same training procedure on all datasets, varying only the batch size. If not noted otherwise, we present results on the separate validation trajectories for Open Catalyst datasets. To simplify the discussion we primarily focus on force mean absolute error (MAE), and show changes as relative improvements compared to the GemNet-OC baseline $\frac{\mathrm{MAE_{GemNet-OC}}}{\mathrm{MAE}} - 1$. We additionally report median relative improvements in throughput (samples per GPU second). We train the baseline model five times for every dataset, but train each ablation only once due to computational restrictions. We assume that the ablations have the same standard deviation as the baseline, and show uncertainties based on the 68 % confidence intervals, calculated using the student's t-distribution for the baseline mean, the standard deviation for the ablations, and appropriate propagation of uncertainty. For further details see App. A.

**Model width and depth.** In general, we would expect larger models to perform better since all investigated settings lie well inside the over-parametrization regime (Belkin et al., 2019). However, the edge embedding size (model width) and the number of blocks (depth) in Fig. 3 actually exhibit optima for most datasets instead of a strictly increasing trend. Importantly and perhaps unsurprisingly, these optima differ between datasets. MD17 exhibits a clear optimum at a low width and depth, and also COLL, OC-200k and OC-sub30 exhibit shallow optima at low width. These optima appear to be mostly present for datasets with a low number of samples compared to its high data complexity and a dissimilar validation set (see Fig. 11 for results on the OOD validation set). Accordingly, we observe the greatest benefit from model size for the largest datasets and in-distribution molecules. We observe a similar effect for other embedding sizes such as the projection size of basis functions. Increasing the projection size yields minor improvements only for OC20 (see Fig. 10). Overall, we notice that increasing the model size further only leads to modest accuracy gains, especially for OOD data. This suggest that molecular machine learning cannot be solved by scaling models alone. Note that this result might be impacted by model initialization (Yang et al., 2021).

**Batch size.** In Fig. 3 we see that most datasets exhibit optima around a batch size of 32 to 128. However, there are substantial differences between datasets. MD17 has a particularly low optimal batch size, as do OOD validation sets (see Fig. 11). MD17 and OOD sets might thus benefit from the regularization effect caused by a small batch size. We also observed that convergence speed is fairly consistent across batch sizes if we focus on the number of samples seen during training, not the number of gradient steps.

**Neighbors and cutoffs.** The first two sections of Fig. 4 shows the impact of the cutoff, the number of neighbors and the number of quadruplets. Their trends are mostly consistent between datasets, albeit with

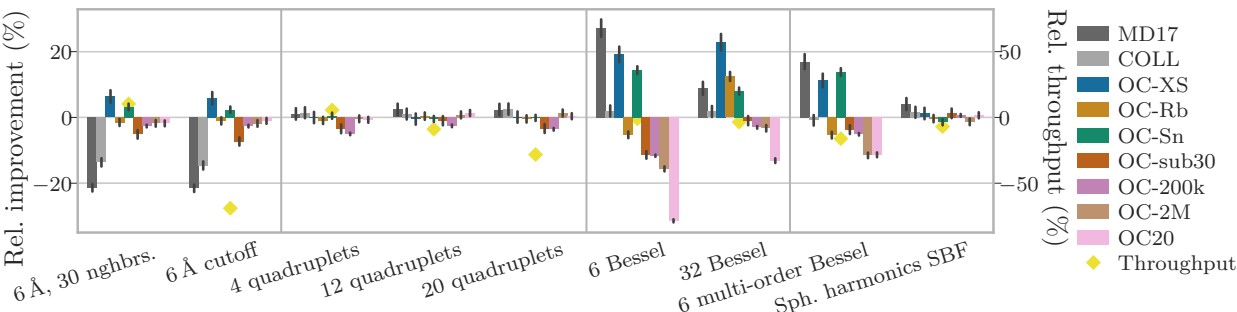

Figure 4: Impact of cutoffs, neighbors, and basis functions on force MAE and training throughput compared to the baseline using a cutoff of 12 Å, 30 neighbors, 8 quadruplets, Legendre polynomials as angular, and Gaussians as radial basis. Notably, the radial Bessel basis performs significantly better on datasets with small chemical diversity, but worse on full OC20.

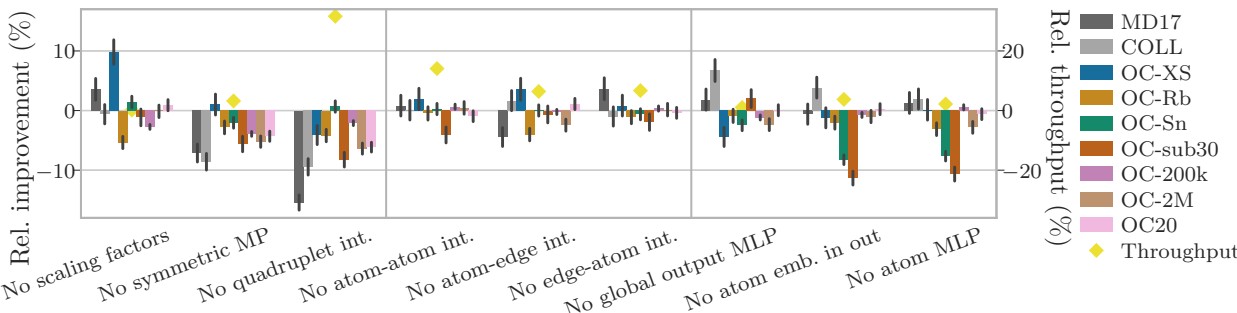

Figure 5: Ablation studies of various components proposed by GemNet and GemNet-OC for force MAE and throughput. Some model changes show consistent improvements (e.g. the quadruplet interaction), while others have very different effects for different datasets.

wildly different impact strengths. For the small molecules in MD17 and COLL the large default cutoff causes a fully-connected interatomic graph, which performs substantially better. Importantly, the default nearest neighbor-based graph performs better than cutoffs on OC20. The number of quadruplets has a surprisingly small impact on model performance on all datasets. We observe a small improvement only for 20 quadruplets, which is likely not worth the associated 20 % runtime overhead.

**Basis functions.** An intruiging trend emerges when comparing 6 and 32 Bessel functions with the default 128 Gaussian basis functions (third section of Fig. 4): Bessel functions perform *substantially* better on datasets with small chemical diversity, such as MD17, OC-XS, OC-Rb, and OC-Sn. However, this trend completely *reverses* on large datasets such as OC20. Multi-order Bessel functions also fit within this trend, as shown in the fourth section of Fig. 4. 6 multi-order Bessel functions perform somewhere between 6 and 32 Bessel functions, since they provide more information than regular 0-order Bessel functions. However, they are significantly slower and thus not recommended. Simplifying the spherical basis part has a similar effect: The outer product of Legendre polynomials is faster and works as well as spherical Harmonics across all datasets.

**Ablation studies.** In Fig. 5, we ablate various components from GemNet-OC to show their effect on different datasets, based both on previous and newly proposed components. This is perhaps the most striking result: Many model components have *opposite* effects on different datasets. The empirical scaling factors proposed by Gasteiger et al. (2021) have a very inconsistent effect on different datasets. Note that we found that they stabilize early training, which can be essential for training with mixed precision even if they do not improve final performance. Symmetric message passing or quadruplet interactions have the most

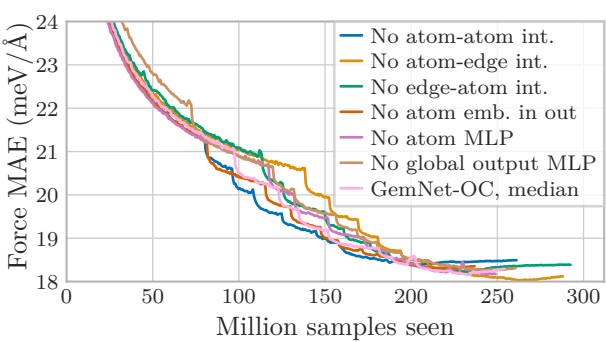

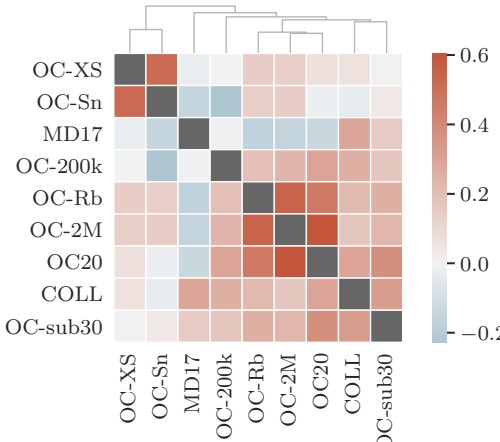

Figure 6: Convergence of model ablations for OC20. Atom-edge and edge-atom interactions, and the global output MLP improve convergence, while atom-atom interactions, the atom MLP, and atom embeddings in the output have no positive effect. Note that step-like improvements are due to adaptive learning rate steps.

Figure 7: Kendall rank correlation of model force MAEs between datasets and resulting hierarchical clustering. Results vary strongly between datasets. OC-2M gives the most similar results to OC20.

consistent impact across datasets. The additional interactions in GemNet-OC show little to no impact on in-distribution OC20, but are beneficial for OC20 out-of-distribution data (see Fig. 13). The atom-to-edge and edge-to-atom interactions furthermore lead to faster convergence, as shown in Fig. 6. Importantly, the impact of these interactions is different on each dataset, varying between significantly negative to significantly positive results. Similarly, the proposed architectural improvements usually have either no or a small positive effect at convergence, but drastically hurt performance on COLL. Predicting forces via backpropagation is another model choice that works well on small datasets (MD17, COLL), but does not benefit the full OC20 dataset. Unfortunately, training GemNet-OC on OC20 with backpropagated forces is too unstable to train reliably and report here.

**Correlation between datasets.** To quantify the overall similarity of model performance between datasets we calculate the Kendall rank correlation coefficient of the force MAEs on one dataset to each other dataset. Each model variant represents one data point in this calculation. We exclude the batch size experiments since this result seems too obvious. We additionally show a hierarchical clustering based on the nearest point algorithm. Fig. 7 shows that OC-2M provides the most similar results to OC20. Interestingly, OC-2M is roughly as similar to OC20 as early stopping or the OOD test set (see Fig. 9). This good correlation is due to OC-2M capturing the same chemical diversity and test difficulty thanks to uniform sampling. It is not an outlier in this respect: We independently sampled five 2M subsets and observed a standard deviation in GemNet-OC force MAE of merely 0.9 %. We furthermore observe that MD17 is especially different from OC20. The OOD validation set (Fig. 14) shows similar results. Note that these aggregate results hide the fact that some model choices, such as using quadruplets, are very consistent between datasets, while others are very different, such as the choice of radial basis.

Based on these results we conclude that a dataset should cover the same chemical diversity to be reflective of a large and diverse dataset. Both the less diverse systems in OC-Rb and OC-Sn and the smaller systems in OC-sub30 show a larger difference in model choices. And these differences add up: Results on OC-XS are even more different. We can preserve this diversity by uniformly subsampling the dataset (OC-2M), but this still breaks down if we reduce the dataset too much (OC-200k). The choice of test set also seems to play a critical role. For example, the OC20 OOD validation set is only slightly closer to in-distribution than OC-2M, and the OC-XS "same trajectory" validation set is very different to regular OC-XS. Note that domain shift appears to make tasks particularly difficult. This is indicated by the (absolute) force MAE: 99.1 meV/Å for OOD on OC-XS, 16.0 meV/Å for separate trajectories (regular validation set), and 3.2 meV/Å for same

trajectories. Overall, it appears that correlated model choices require datasets to have comparable difficulty and complexity, as given by chemical diversity, domain shift, and sufficient training set size.

## 7 Conclusion

In this work we studied the effect of developing GNNs on a large dataset instead of small benchmarks. We proposed the GemNet-OC model, which is substantially faster to train than previous models and sets the state of the art on all OC20 tasks. We then investigated how model choices in GemNet-OC differ between datasets. We found that model components and hyperparameters can have disparate or even opposite effects between datasets, even within the single task of force predictions. We studied the source of this effect by selecting data subsets that individually change four main data complexity aspects. This resulted in two insights: First, consistent model choices require datasets with comparable difficulty and complexity, as given by their chemical diversity, domain shift, and a sufficiently large training set. Second, we can create a well-correlated proxy dataset by uniformly sampling a sufficiently large data subset. The resulting OC-2M dataset allows model development for the massive OC20 dataset at a fraction of the computational cost. To support future model research we provide a range of baseline results for this dataset. Overall, this case study highlights that researchers should exercise caution during model development, since model choices can vary drastically when changing relevant dataset properties.

## Acknowledgments

We thank Brandon Wood for helpful discussions on performance and scaling aspects and the Open Catalyst team for their support, feedback, and discussions and for providing the foundational codebase for this project (Chanussot et al., 2021). We furthermore acknowledge PyTorch (Paszke et al., 2019) and PyTorch Geometric (Fey & Lenssen, 2019).

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

# A    Training and hyperparameters

All dataset and model ablations used the same base model hyperparameters detailed in Table 3. Base effective batch sizes (see Table 4) varied across dataset ablations in order to reach convergence in a reasonable amount of time across the large dataset size differences. MD17 and COLL ablations used a learning rate scheduler consistent with that of Gasteiger et al. (2021) - linear warmup, exponential decay, and reduce on plateau. OC ablations only used reduce on plateau, with evaluations performed after a fixed number of steps (1k or 5k) instead of after each epoch. MD17, COLL, OC-Rb, OC-XS, and OC-200k were trained on 40GB NVIDIA A100 cards with the remaining datasets trained on 32GB NVIDIA V100 cards. We provide the hyperparameters for our best performing model variant, GemNet-OC-Large in Table 3.

OC energies were standardized by subtracting the mean energy of the dataset and dividing by the standard deviation. Forces were divided by the same energy standard deviation in order to ensure consistency with their physical relationship: $F = \frac{-dE}{dx}$. For MD17 and COLL, only the mean energy was subtracted Gasteiger et al. (2021). All models were trained within the ocp repository and optimized to the following loss function (1)

$$
\begin{aligned}
L(\mathbf{X}, z) = \lambda \left| f_\theta(\mathbf{X}, \mathbf{z}) - \hat{E}(\mathbf{X}, \mathbf{z}) \right| \\
+ \frac{\rho}{N} \sum_{n=1}^{N} \sqrt{\sum_{\alpha=1}^{3} \left( g_{\theta,n\alpha}(\mathbf{X}, \mathbf{z}) - \hat{F}_{n\alpha}(\mathbf{X}, \mathbf{z}) \right)^2},
\end{aligned}
\tag{1}
$$

where $\lambda$ corresponds to the energy coefficient, $\rho$ corresponds to the force coefficient, $\mathbf{X}$ are the atom coordinates, $\mathbf{z}$ are the atomic numbers, $f_\theta$ and $\mathbf{g}_\theta$ are learnable functions with shared parameters $\theta$, $\hat{E}$ is the ground-truth energy, and $\hat{\mathbf{F}}$ are the ground-truth forces. All model ablations make direct force predictions. Although models on the MD17 and COLL datasets generally perform better with gradient-based force predictions $g_{\theta,n\alpha}(\mathbf{X}, \mathbf{z}) = \frac{\partial f_\theta(\mathbf{X}, \mathbf{z})}{\partial x_{n\alpha}}$, the same is not true for OC20 datasets. Gradient-based models on this dataset often run into numerical instabilities and hit NaNs very early in training. All models were optimized using AMSGrad Reddi et al. (2018) and trained for a max number of epochs (4) or until the learning rate has been exhaustively stepped, whichever comes first.

# B    Additional experimental results

**Early stopping.** To obtain model results faster one might be tempted to stop model training early and draw conclusions from early model performance. However, as shown in Fig. 6, early results can be misleading and often do not reflect final performance. Fig. 8 shows that early stopping also has a strong effect on the choice of basis function, similar to chemical diversity. While a large Bessel basis works substantially better early in training, this trend *reverses* when the model approaches convergence. Overall, results only become well-correlated with final OC20 accuracy late in training (see Fig. 9. Stopping training early should thus not be considered as a way of accelerating model development).

**Complementary results.** Complementary to the results in the main paper, Figs. 11 to 14 visualize similar plots across the out-of-distribution splits of the proposed datasets. Tables 5 to 8 separately show the test results for each OC20 test split.

Table 3: Model hyperparameters for the baseline configuration and our GemNet-OC-Large model.

| Hyperparameters | Base | GemNet-OC-Large |
|---|---|---|
| No. spherical basis | 7 | 7 |
| No. radial basis | 128 | 128 |
| No. blocks | 4 | 6 |
| Atom embedding size | 256 | 256 |
| Edge embedding size | 512 | 1024 |
| | | |
| Triplet edge embedding input size | 64 | 64 |
| Triplet edge embedding output size | 64 | 128 |
| Quadruplet edge embedding input size | 32 | 64 |
| Quadruplet edge embedding output size | 32 | 32 |
| Atom interaction embedding input size | 64 | 64 |
| Atom interaction embedding output size | 64 | 64 |
| Radial basis embedding size | 16 | 32 |
| Circular basis embedding size | 16 | 16 |
| Spherical basis embedding size | 32 | 64 |
| | | |
| No. residual blocks before skip connection | 2 | 2 |
| No. residual blocks after skip connection | 2 | 2 |
| No. residual blocks after concatenation | 1 | 4 |
| No. residual blocks in atom embedding blocks | 3 | 3 |
| No. atom embedding output layers | 3 | 3 |
| | | |
| Cutoff | 12.0 | 12.0 |
| Quadruplet cutoff | 12.0 | 12.0 |
| Atom edge interaction cutoff | 12.0 | 12.0 |
| Atom interaction cutoff | 12.0 | 12.0 |
| Max interaction neighbors | 30 | 30 |
| Max quadruplet interaction neighbors | 8 | 8 |
| Max atom edge interaction neighbors | 20 | 20 |
| Max atom interaction neighbors | 1000 | 1000 |
| | | |
| Radial basis function | Gaussian | Gaussian |
| Circular basis function | Spherical harmonics | Spherical harmonics |
| Spherical basis function | Legendre Outer | Legendre Outer |
| Quadruplet interaction | True | True |
| Atom edge interaction | True | True |
| Edge atom interaction | True | True |
| Atom interaction | True | True |
| Direct forces | True | True |
| | | |
| Activation | Silu | Silu |
| Optimizer | AdamW | AdamW |
| Force coefficient | 100 | 100 |
| Energy coefficient | 1 | 1 |
| EMA decay | 0.999 | 0.999 |
| Gradient clip norm threshold | 10 | 10 |
| Learning rate | 0.001 | 0.0005 |

Table 4: Training hyperparameters across all proposed datasets.

|  | OC-20 | OC-2M | OC-200k | OC-sub30 | OC-XS | OC-Rb | OC-Sn | MD17 | COLL |
|---|---|---|---|---|---|---|---|---|---|
| Effective batch size | 128 | 64 | 64 | 64 | 16 | 64 | 16 | 1 | 32 |
| No. GPUs | 16 | 8 | 4 | 2 | 1 | 4 | 2 | 1 | 1 |
| Max epochs | 80 | 80 | 50 | 50 | 100 | 100 | 100 | 10000 | 1000 |

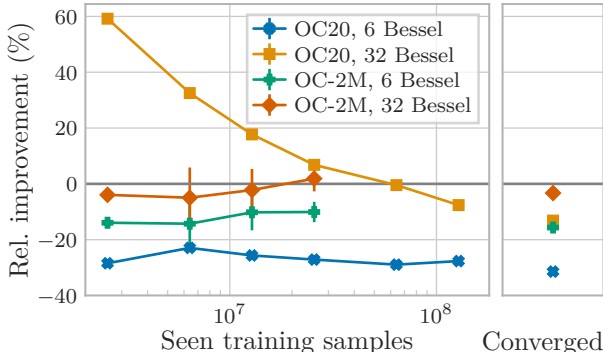

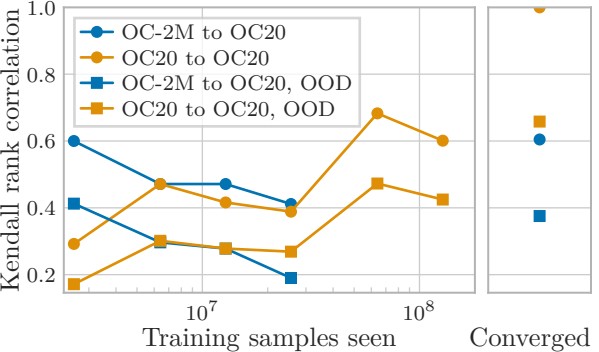

Figure 8: Effect of radial basis functions during training on force MAE. Bessel functions work best on OC20 early in training, but this trend reverses at convergence. OC-2M exhibits a more consistent behavior.

Figure 9: Kendall rank correlation of model force MAEs during training to the final result. The correlation shows a drop early in training, which is likely due to the variance caused by the learning rate (LR) decay on plateau schedule. Correlation then increases again during late training. We found that LR variance has no impact on final converged results. Note that the converged OC-2M points have seen 56 M samples on average, putting it roughly on par with early stopping.

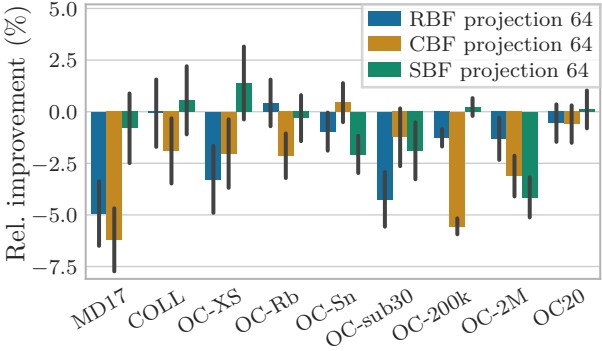

Figure 10: Impact of basis projection sizes on force MAE. Large projection sizes have a minor beneficial effect on OC20, while being detrimental on almost all smaller datasets.

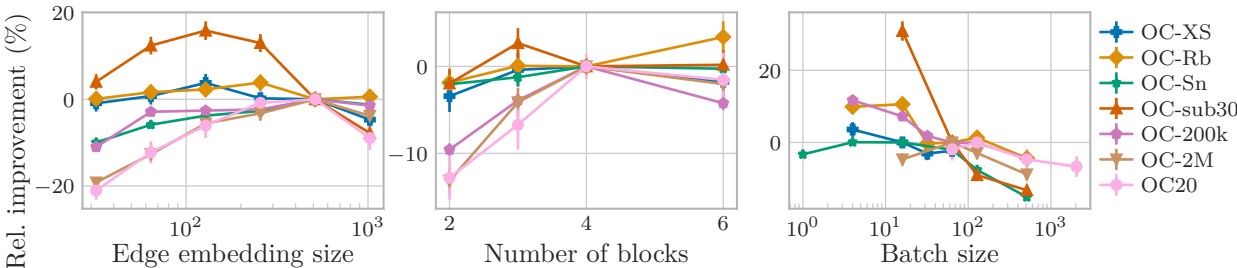

Figure 11: Effect of batch size, edge embedding size, and number of blocks on force MAE for the OOD validation set. Model size optima emerge on most datasets as we move to the OOD dataset. Note that OC-XS, OC-Rb, and OC-Sn use in-distribution catalysts.

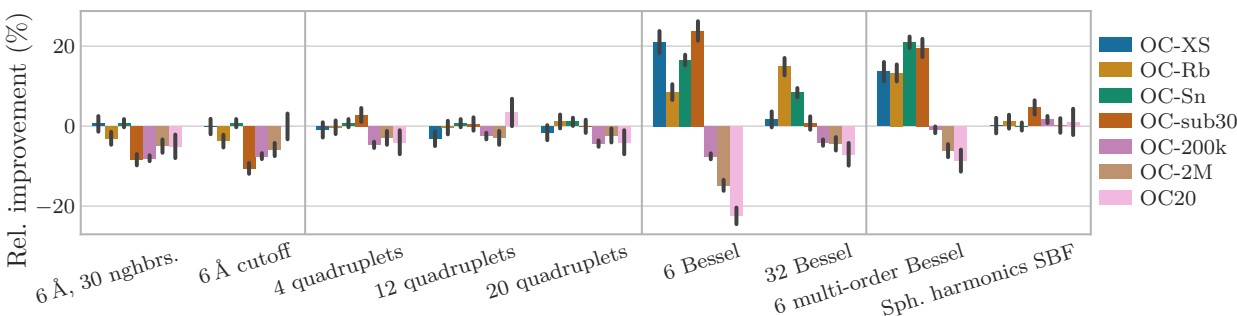

Figure 12: Impact of cutoffs, neighbors, and basis functions on force MAE for the OOD validation set. The Bessel basis performs signficantly better on datasets with small chemical diversity, but worse on OC20 - consistent with the in-distribution trends.

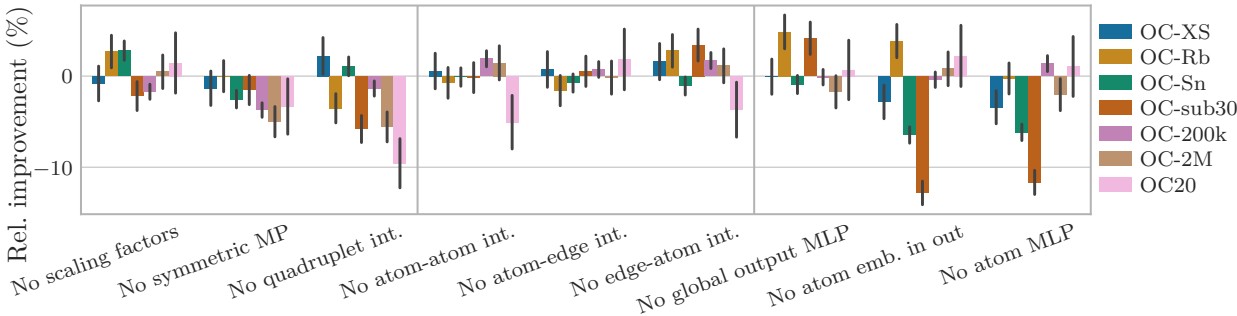

Figure 13: Ablation studies for various components proposed by GemNet and GemNet-OC, based on force MAE on the OOD validation set. Some model changes are fairly consistent across OC20 datsets (e.g. symmetric message passing, quadruplet interaction), while others have varying effects across datasets.

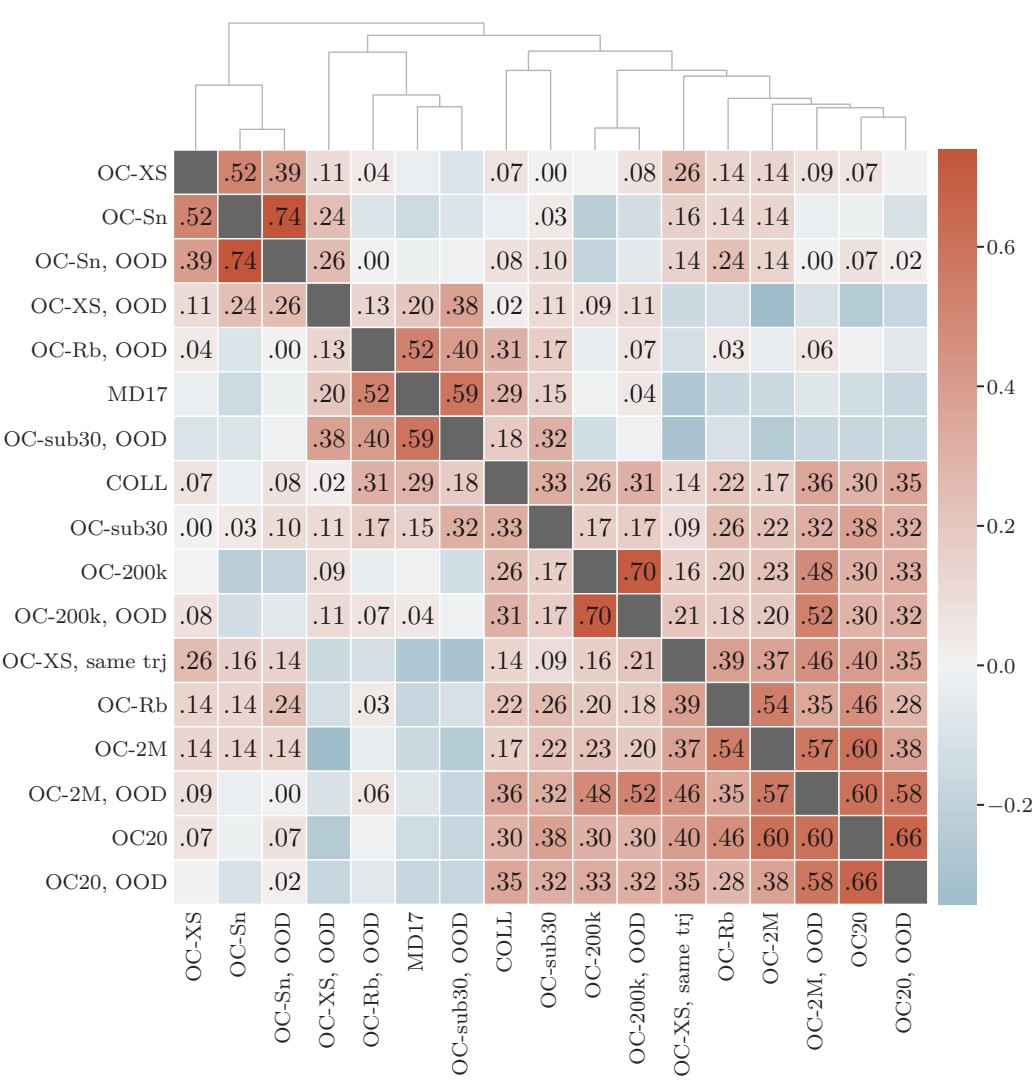

Figure 14: Kendall rank correlation of model force MAEs and resulting hierarchical clustering of datasets and different validation sets. Positive correlations are annotated. OC-2M is the most correlated to OC20, both in- and out-of-distribution.

Table 5: Results for the OC20 in-distribution (ID, also called "separate trajectories") validation set and the three test tasks.

| Train set | Model | S2EF validation | | | | S2EF test | | | | IS2RS | | IS2RE |
|---|---|---|---|---|---|---|---|---|---|---|---|---|
| | | Energy MAE meV ↓ | Force MAE meV/Å ↓ | Force cos ↑ | EFwT % ↑ | Energy MAE meV ↓ | Force MAE meV/Å ↓ | Force cos ↑ | EFwT % ↑ | AFbT % ↑ | ADwT % ↑ | Energy MAE meV ↓ |
| OC-2M | SchNet | 1360 | 73.7 | 0.112 | 0.00 | 1370 | 73.6 | 0.117 | 0.00 | - | - | - |
| | DimeNet++ | 737 | 59.2 | 0.229 | 0.02 | 738 | 59.2 | 0.227 | 0.01 | - | - | - |
| | SpinConv | 360 | 32.8 | 0.485 | 0.23 | 357 | 32.7 | 0.479 | 0.22 | - | - | - |
| | GemNet-dT | 275 | 25.5 | 0.574 | 1.15 | 271 | 25.5 | 0.571 | 1.19 | 20.0 | 55.1 | 435 |
| | GemNet-OC | **226** | **22.5** | **0.610** | **1.89** | **226** | **22.5** | **0.610** | **1.94** | **22.4** | **56.5** | **405** |
| OC20 | CGCNN | 504 | 68.4 | 0.155 | 0.01 | 511 | 68.3 | 0.154 | 0.01 | - | - | - |
| | SchNet | 447 | 49.3 | 0.319 | 0.13 | 442 | 49.3 | 0.318 | 0.11 | - | 15.2 | 709 |
| | ForceNet-large | - | 28.1 | 0.534 | - | - | 31.2 | 0.520 | 0.01 | 14.8 | 50.6 | - |
| | DimeNet++-L-F+E | 360 | 28.1 | 0.564 | 0.00 | 359 | 28.0 | 0.564 | 0.00 | 25.6 | 52.4 | 502 |
| | PaiNN | - | - | - | - | 248 | 29.3 | 0.511 | 0.88 | 15.6 | 49.5 | 442 |
| | SpinConv | 287 | 37.0 | 0.479 | 0.09 | 261 | 26.9 | 0.548 | 0.82 | 21.1 | 53.7 | 424 |
| | GemNet-dT | 242 | 22.8 | 0.613 | 1.13 | 226 | 21.0 | 0.637 | 2.40 | 33.8 | 59.2 | 390 |
| | GemNet-XL | - | - | - | - | 212 | 18.1 | 0.676 | 3.30 | 34.6 | **62.7** | 376 |
| | GemNet-OC | **172** | **17.9** | **0.685** | **4.59** | **168** | **17.9** | **0.686** | **4.70** | **40.7** | 60.6 | **348** |
| OC20+ OC-MD | GemNet-OC-L-E | **153** | 17.8 | 0.688 | 5.30 | **150** | 17.8 | 0.688 | 5.44 | - | - | - |
| | GemNet-OC-L-F | 170 | **16.3** | **0.711** | **5.35** | 163 | **16.3** | **0.711** | **5.47** | 47.4 | 60.8 | - |
| | GemNet-OC-L-F+E | - | - | - | - | - | - | - | - | - | - | **331** |

Table 6: Results for the OC20 out-of-distribution adsorbates validation set and the three test tasks.

| Train set | Model | S2EF validation | | | | S2EF test | | | | IS2RS | | IS2RE |
|---|---|---|---|---|---|---|---|---|---|---|---|---|
| | | Energy MAE meV ↓ | Force MAE meV/Å ↓ | Force cos ↑ | EFwT % ↑ | Energy MAE meV ↓ | Force MAE meV/Å ↓ | Force cos ↑ | EFwT % ↑ | AFbT % ↑ | ADwT % ↑ | Energy MAE meV ↓ |
| OC-2M | SchNet | 1410 | 77.3 | 0.108 | 0.00 | 1340 | 74.1 | 0.114 | 0.00 | - | - | - |
| | DimeNet++ | 806 | 67.0 | 0.203 | 0.00 | 694 | 61.0 | 0.215 | 0.01 | - | - | - |
| | SpinConv | 375 | 35.6 | 0.479 | 0.05 | 350 | 33.7 | 0.466 | 0.09 | - | - | - |
| | GemNet-dT | 309 | 29.3 | 0.560 | 0.21 | 269 | 26.5 | 0.557 | 0.59 | 15.9 | 50.5 | 442 |
| | GemNet-OC | **258** | **25.2** | **0.600** | **0.45** | **235** | **22.9** | **0.597** | **1.09** | **19.5** | **52.2** | **416** |
| OC20 | CGCNN | 599 | 74.6 | 0.132 | 0.00 | 632 | 72.8 | 0.137 | 0.00 | - | - | - |
| | SchNet | 497 | 57.4 | 0.286 | 0.00 | 486 | 52.9 | 0.295 | 0.04 | - | 12.8 | 774 |
| | ForceNet-large | - | 32.0 | 0.520 | - | - | 28.3 | 0.521 | 0.01 | 12.2 | 45.2 | - |
| | DimeNet++-L-F+E | 450 | 31.8 | 0.550 | 0.00 | 402 | 28.9 | 0.550 | 0.00 | 20.7 | 48.5 | 543 |
| | PaiNN | - | - | - | - | 280 | 30.0 | 0.499 | 0.43 | 12.2 | 44.3 | 480 |
| | SpinConv | 314 | 40.0 | 0.471 | 0.03 | 275 | 27.7 | 0.535 | 0.38 | 15.7 | 48.9 | 442 |
| | GemNet-dT | 247 | 25.4 | 0.605 | 0.30 | 210 | 21.9 | 0.624 | 1.15 | 26.8 | 54.6 | 391 |
| | GemNet-XL | - | - | - | - | 198 | 18.6 | 0.664 | 1.62 | 30.3 | **58.6** | 368 |
| | GemNet-OC | **189** | **19.6** | **0.681** | **1.27** | **171** | **18.2** | **0.674** | **2.57** | **36.1** | 56.6 | **350** |
| OC20+ OC-MD | GemNet-OC-L-E | **178** | 19.5 | 0.685 | 1.66 | **152** | 18.1 | 0.678 | **3.21** | - | - | - |
| | GemNet-OC-L-F | 195 | **17.9** | **0.707** | **1.70** | 174 | **16.6** | **0.700** | 3.18 | 40.4 | 56.4 | - |
| | GemNet-OC-L-F+E | - | - | - | - | - | - | - | - | - | - | **336** |

Table 7: Results for the OC20 out-of-distribution catalysts validation set and the three test tasks.

| Train set | Model | S2EF validation Energy MAE meV ↓ | Force MAE meV/Å ↓ | Force cos ↑ | EFwT % ↑ | S2EF test Energy MAE meV ↓ | Force MAE meV/Å ↓ | Force cos ↑ | EFwT % ↑ | IS2RS AFbT % ↑ | ADwT % ↑ | IS2RE Energy MAE meV ↓ |
|---|---|---|---|---|---|---|---|---|---|---|---|---|
| OC-2M | SchNet | 1330 | 72.7 | 0.105 | 0.00 | 1340 | 71.7 | 0.114 | 0.00 | - | - | - |
| | DimeNet++ | 738 | 58.9 | 0.223 | 0.02 | 745 | 58.0 | 0.219 | 0.01 | - | - | - |
| | SpinConv | 399 | 33.6 | 0.461 | 0.20 | 398 | 33.5 | 0.460 | 0.16 | - | - | - |
| | GemNet-dT | 374 | 27.3 | 0.533 | 0.97 | 341 | 26.8 | 0.536 | 0.76 | 15.3 | 54.8 | 454 |
| | GemNet-OC | **288** | **24.0** | **0.576** | **1.68** | **279** | **23.3** | **0.584** | **1.44** | **19.8** | **56.7** | **416** |
| OC20 | CGCNN | 525 | 67.9 | 0.146 | 0.00 | 520 | 67.0 | 0.149 | 0.01 | - | - | - |
| | SchNet | 545 | 52.0 | 0.297 | 0.10 | 528 | 50.9 | 0.296 | 0.06 | - | 14.6 | 767 |
| | ForceNet-large | - | 32.7 | 0.491 | - | - | 30.9 | 0.494 | 0.01 | 12.2 | 49.8 | - |
| | DimeNet++-L-F+E | 541 | 31.5 | 0.511 | 0.00 | 504 | 31.2 | 0.512 | 0.00 | 20.1 | 50.9 | 578 |
| | PaiNN | - | - | - | - | 366 | 32.7 | 0.460 | 0.40 | 8.93 | 47.8 | 486 |
| | SpinConv | 397 | 39.7 | 0.454 | 0.05 | 350 | 28.5 | 0.519 | 0.46 | 15.9 | 53.9 | 457 |
| | GemNet-dT | 357 | 26.9 | 0.561 | 0.64 | 340 | 24.5 | 0.581 | 0.93 | 24.7 | 58.7 | 434 |
| | GemNet-XL | - | - | - | - | 308 | **20.6** | 0.631 | 1.72 | 29.3 | **62.6** | 402 |
| | GemNet-OC | **272** | **22.3** | **0.621** | **2.08** | **266** | 21.4 | **0.632** | **1.95** | **33.0** | 60.6 | **377** |
| OC20+ OC-MD | GemNet-OC-L-E | **278** | 23.1 | 0.617 | 1.86 | **282** | 22.1 | 0.628 | 1.82 | - | - | - |
| | GemNet-OC-L-F | 287 | **20.7** | **0.646** | **2.51** | 284 | **20.0** | **0.656** | **2.29** | 37.4 | 60.8 | - |
| | GemNet-OC-L-F+E | - | - | - | - | - | - | - | - | - | - | **379** |

Table 8: Results for the OC20 out-of-distribution both (adsorbate and catalyst) validation set and the three test tasks.

| Train set | Model | S2EF validation Energy MAE meV ↓ | Force MAE meV/Å ↓ | Force cos ↑ | EFwT % ↑ | S2EF test Energy MAE meV ↓ | Force MAE meV/Å ↓ | Force cos ↑ | EFwT % ↑ | IS2RS AFbT % ↑ | ADwT % ↑ | IS2RE Energy MAE meV ↓ |
|---|---|---|---|---|---|---|---|---|---|---|---|---|
| OC-2M | SchNet | 1490 | 89.6 | 0.110 | 0.00 | 1440 | 89.1 | 0.117 | 0.00 | - | - | - |
| | DimeNet++ | 940 | 77.5 | 0.214 | 0.00 | 867 | 73.7 | 0.226 | 0.00 | - | - | - |
| | SpinConv | 492 | 42.8 | 0.492 | 0.03 | 500 | 42.1 | 0.494 | 0.05 | - | - | - |
| | GemNet-dT | 475 | 36.0 | 0.559 | 0.11 | 413 | 33.5 | 0.573 | 0.23 | 15.7 | 58.8 | 420 |
| | GemNet-OC | **370** | **31.0** | **0.606** | **0.23** | **355** | **28.5** | **0.620** | **0.52** | **16.6** | **60.3** | **391** |
| OC20 | CGCNN | 731 | 85.2 | 0.134 | 0.01 | 768 | 85.1 | 0.144 | 0.00 | - | - | - |
| | SchNet | 705 | 68.5 | 0.285 | 0.00 | 706 | 65.5 | 0.299 | 0.01 | - | 14.8 | 806 |
| | ForceNet-large | - | 41.2 | 0.516 | - | - | 37.5 | 0.530 | 0.00 | 11.5 | 52.9 | - |
| | DimeNet++-L-F+E | 711 | 39.6 | 0.539 | 0.00 | 655 | 37.1 | 0.552 | 0.00 | 20.6 | 54.9 | 612 |
| | PaiNN | - | - | - | - | 470 | 40.5 | 0.493 | 0.13 | 10.1 | 52.2 | 474 |
| | SpinConv | 487 | 48.2 | 0.486 | 0.01 | 459 | 35.6 | 0.555 | 0.14 | 14.0 | 58.0 | 425 |
| | GemNet-dT | 415 | 33.5 | 0.596 | 0.10 | 394 | 29.6 | 0.622 | 0.30 | 25.1 | 62.2 | 384 |
| | GemNet-XL | - | - | - | - | 362 | **24.5** | 0.670 | 0.61 | 29.0 | **66.7** | **338** |
| | GemNet-OC | **344** | **27.1** | **0.659** | **0.36** | **326** | 25.2 | **0.671** | **0.79** | **31.3** | 63.5 | 342 |
| OC20+ OC-MD | GemNet-OC-L-E | **347** | 27.9 | 0.658 | 0.38 | **336** | 25.9 | 0.669 | 0.74 | - | - | - |
| | GemNet-OC-L-F | 357 | **25.1** | **0.685** | **0.50** | 343 | **23.1** | **0.696** | **0.93** | 37.1 | 63.7 | - |
| | GemNet-OC-L-F+E | - | - | - | - | - | - | - | - | - | - | **344** |

