# OpenReview forum: "GemNet-OC: Developing Graph Neural Networks for Large and Diverse Molecular Simulation Datasets"
_TMLR — Accepted by TMLR_

### Review · Reviewer_PhJC · 2022-07-25

**Summary Of Contributions:**

This work performs a comprehensive (empirical) study for GNNs over molecular simulation datasets. It first proposes GemNet-OC by improving the previous work GemNet. Then, comprehensive empirical analyses have been conducted to show that model components and hyperparameters can have disparate or even opposite effects between datasets. To alleviate this issue and facilitate future research, a refined dataset OC-2M has been developed and empirically verified as a convincing benchmark.

**Requested Changes:**

(1) It would be clearer to the community if the consumed computational cost is provided.

**Strengths And Weaknesses:**


Strengths:

(1) Most studies and observations of this work are based on empirical results. These comprehensive results could be useful for the community.

(2) The presentation of this work is great. The observations are clearly summarized and explained, with well-organized empirical results.

(3) The improvements of GemNet-OC over the original GemNet are obvious. These designs are helpful for the community.

Weaknesses:

(1) Since this work is mainly empirical studies, the methodology-wise novelty is not strong. Having said this, the value of these empirical observations is still great.

(2) To derive the comprehensive experiments in this work, I expect that the computational cost is quite high. It would be better to summarize how much computational cost have been used to obtain these results.

---

> ### Author Response · Authors · 2022-08-24
> **Rigorous empiricism & computational resources**
>
> Thank you for the precise review! We are happy to hear that you find our results comprehensive and well-organized, our presentation great and clear, and our improvements obvious and helpful!
>
> The focus of our paper is indeed a thorough empirical analysis with clearly stated, precise results. To our understanding, TMLR is meant to emphasize exactly these aspects of rigor.
>
> You are correct in assuming that we have invested large computational resources to generate these studies and results. In total, the presented  analyses and models used roughly 16000 GPU days for training on Nvidia V100 32GB GPUs. We have added this information to the paper. Note that these resources were spent so others do not have to. One main goal of our thorough analysis is allowing future researchers to take shortcuts and produce generalizable results with fewer resources.

---

> > ### Comment · Reviewer_PhJC · 2022-08-29
> > **Thanks**
> >
> > Thank you for the clarification.

---

### Review · Reviewer_AHac · 2022-07-26

**Summary Of Contributions:**

This paper investigates the GemNet model developed for the Open Catalyst benchmark. It looks at various aspects of the dataset in conjunction with details of the model, coming up with several findings: (1) smaller and less diverse datasets than OC are a bad proxy for model development for OC, (2) a random subsample of OC is a good proxy, and (3) GemNet's performance can be improved significantly if one is directly optimizing for OC. Overall, the paper pushes SotA on OC20, as well as provides many guidelines and interesting observations for developing models on molecular datasets.

**Broader Impact Concerns:**

I have no ethical concerns about the impact of this work. Better models can of course be used for good or bad, but this is expected and not specific to this paper. If anything, this work may reduce the biases in modelling choices (by discouraging transferring from small toy-ish datasets to challenging ones), which should have a good influence on the community.

**Requested Changes:**

I think the paper is in good shape for publication, and I don't have significant changes to request. If the authors have any interesting thoughts regarding my questions, it would be nice to include these in the manuscript, otherwise it's fine to skip them. It would also be helpful to address the nitpicks, but this is just to slightly polish the paper and not a critical change.

**Strengths And Weaknesses:**

=== Strengths ===

- (S1): The paper takes a very scientific approach to comparing different benchmarking datasets, taking them apart from multiple angles, and running many comparisons. While many of the final results aren't wildly surprising (e.g. a subsample of a large and diverse dataset is a better proxy for it than a different small and less diverse dataset), they are backed by a rigorous quantitative study.

- (S2): Through careful design decisions, the authors push SotA on OC through their GemNet-OC model. The model modifications are often relatively small details, but again, the choices are backed by good experiments and analysis.

- (S3): The paper is very well written, with practically zero typos, grammar errors or confusing sentences. The text is easy to read, and it gets the main points across rather precisely. Writing quality is better than the majority of papers accepted at NeurIPS/ICML/ICLR.



=== Weaknesses ===

- (W1): The paper presents a large-scale but purely empirical study, and one could say the scientific novelty is limited. However, I would say this is not an important weakness; I much prefer a paper presenting a good deal of quantitative insights (as this one), instead of novel ideas with unclear applicability or significance.



=== Questions ===

- (Q1): I haven't been closely following the recent SotA improvement on OC20, and thus I wonder how to contextualize Table 2 with regards to the original challenge. Am I correct in thinking that the OC20 challenge winner (Graphormer model from MSR Asia) is not in the table? I understand the recent models there surpassed the original Graphormer result significantly, but maybe it would still make sense to have it there to connect to the original challenge?

- (Q2): The choice to use a fixed number of neighbors per atom instead of a fixed cutoff is interesting. Did the authors also try using a dynamically chosen cutoff, by computing all atom-atom distances, sorting them, and taking a cutoff such that a constant *average* degree is obtained?

- (Q3): The paper shows that the optimal choice of the basis functions is wildly different depending on the level of chemical diversity in the dataset. Do the authors have any intuition for why this is the case?



=== Nitpicks ===

Here I add some final nitpicks (some are incredibly nitpicky, but as I said the paper is already looking good):

- "In practise we apply a trained model" - as far as I understand, "practise" is only used in British English and only as a verb. Since this here is a noun, perhaps "practice" would be more appropriate.

- Maybe make the color of the word "pink" in the caption of Figure 1 exactly match the color of the pink boxes in the figure?

- Figure 3 appears before Figure 2, maybe swap their numbers.

- "if we focussed on another dataset" - I understand "focussed" is correct albeit uncommon, I wonder if the authors wanted to use "focused" instead.

---

> ### Author Response · Authors · 2022-08-24
> **Addressing your questions**
>
> Thank you for the thorough review and your helpful suggestions! We are glad to hear that you find our approach scientific, our analysis rigorous, and the text easy to read.
>
> **(W1)** The focus of our paper is indeed a thorough empirical analysis with clearly stated, precise results. To our understanding, TMLR is meant to emphasize exactly these aspects of rigor.
>
> **(Q1)** The 3D-Graphormer is a direct IS2RE model, while all presented models are relaxation-based, i.e. they generate predictions based on relaxing the energy of an S2EF model. This is significantly more accurate but also more expensive than direct models. Adding direct models to the table would thus be rather misleading. That being said, we do mention 3D-Graphormer and its MAE (472meV vs. GemNet-OC's 355meV) in the main text.
>
> **(Q2)** We tried multiple variants of the nearest-neighbor graph that would allow for a more continuous cutoff. For example, we tried gradually reducing the weight of neighbors and defining a cutoff based on the N+1'th neighbor and adjusting the envelope function based on this. Unfortunately, these all performed worse than the regular nearest-neighbor graph. The main reason for this is likely that changing the envelope function influences the interactions in ways that the model cannot easily adapt to.
>
> As suggested, we ran an experiment of GemNet-OC with a constant average degree per system. We select the closest atoms without adjusting the cutoff, similar to regular GemNet-OC. If we adjust the cutoff that would impact the envelope function and we'd run into the same issue as above. We found that this variant performs slightly worse than the nearest-neighbor graph. Trained on OC-2M, it achieves an average validation energy MAE of 291meV (vs. 286meV for regular GemNet-OC) and a force MAE of 26.1meV/A (vs. 25.7meV/A). Note that this method loses the benefit of using tensors with a fixed dimension.
>
> **(Q3)** Our main intuition for this is that the radial Bessel basis functions provide a better inductive bias, but do not provide enough channels for learning functions for the full diversity of chemistry. However, we cannot increase the number of channels by increasing the basis size since this also increases the frequency. In contrast, the Gaussian basis functions have less inductive bias but can more easily provide different functions for different atom types. We have tested this hypothesis in subsequent work and indeed found that circumventing this issue by using multiple basis function "modules" does outperform both Bessel and Gaussian basis functions.
>
> **(Nitpicks)** Thank you for pointing these out! We have fixed them in the revised paper.

---

### Review · Reviewer_TH9Z · 2022-08-16

**Summary Of Contributions:**

The work focuses on the molecular simulation problem with the OC20 dataset. The authors propose a variant of GemNet, namely GemNet-OC to improve both efficiency and effectiveness on the OC20 tasks. The authors also conduct comprehensive empirical studies on models’ behaviors and different datasets, and construct a distilled data OC-2M that can further reduce training time and achieve a similar level of performance.

**Requested Changes:**

- The title and paper use the term GNN. In fact the proposed methods as well as baselines are not exactly typical GNNs. I suggest use 3D GNNs or geometric GNNs specifically to avoids confusions with general GNN structures like GCN, GIN and GAT.
- Some recent related 3D GNNs studies are not discussed or compared in the results, e.g., [1], Graphormer [2], SphereNet [3].

[1] Schütt et al., Equivariant message passing for the prediction of tensorial properties and molecular spectra. ICML 2021

[2] Ying et al., Do Transformers Really Perform Badly for Graph Representation? NeurIPS 2021.

[3] Liu et al., Spherical Message Passing for 3D Molecular Graphs. ICLR 2022.

**Strengths And Weaknesses:**

Strengths

- The proposed GemNet variant significantly reduce the training time in terms of GPU hours, without performance loss in general.
- Experimental results are good, indicating a significant improvement from baseline methods including the existing version of GemNet.
- The faster training also enables thorough empirical and ablation study on the impact of individual model components, which provide insights to the model designing of the the problem.
- The constructed subset OC-2M for training can largely reflex the trend of model performance on variety of test sets and metrics. It can be used for development purposes with a much shorter turnaround time, which benefits future studies on the molecular simulation problem.

Weaknesses

- The method novelty and technical contribution are limited. Compared to the original GemNet, the proposed variant only differs in the structures by modifying blocks for efficiency purpose.
- The results are mostly empirical. It is not clear whether the major performance boost comes from the changes in structure or more exhaustive tuning enabled by the faster training (e.g., as shown in Fig 3). It would be better to including more discussions or justifications on the changes.
- It would make more impact if the propose method and empirical results can be generalize to other problems or tasks related to molecule geometry using the 3D GNNs.

---

> ### Author Response · Authors · 2022-08-24
> **Rigorous empiricism, molecular simulation & baselines**
>
> We are happy to hear that you find GemNet-OC's time and accuracy improvements significant, our empirical study insightful for model design, and the proposed OC-2M dataset beneficial for future studies.
>
> ## Changes in GemNet-OC (ad W1)
>
> The focus on computational efficiency in GemNet-OC was deliberate. As stated in Section 1, our paper focuses on accelerating research and studying how different datasets affect model choices. To strengthen this study we stayed relatively close to the original GemNet model and did not incorporate unnecessarily large changes. GemNet-OC is able to accelerate research without sacrificing accuracy and stays comparable to the original GemNet.
>
> ## Empiricism and sources of improvement (ad W2)
>
> We agree that our paper focuses on a thorough empirical analysis with clearly stated, precise results. To our understanding, TMLR is meant to emphasize exactly these aspects of rigor. Moreover, one main question our work is how model trends change across different real-world datasets (Section 6). This question can currently only be answered empirically.
>
> However, we humbly disagree on the point that it is unclear were the major performance boost comes from. We have carefully dissected the model and every hyperparameter in Figures 3, 4, 5, and 6. These figures show precisely and comprehensively which changes cause performance improvements. We did not perform any hyperparameter tuning beyond what is shown in the paper, and all hyperparameters are kept constant in our ablation studies. We invite anyone to verify this by checking the provided configurations and running our code. Our results are fully transparent.
>
> ## Importance of molecular simulation (ad W3)
>
> Molecular simulation, i.e. predicting energies and forces, is probably the most important task in computational chemistry. An accurate and efficient model would be foundational and solve a huge number of tasks and applications, such as predicting reactions, reaction rates, the binding affinity of drugs, or the efficacy of a catalyst. While we agree that extending to other tasks would be interesting, it is beyond the scope of this paper -- especially since even across datasets for this task, there is significant variance in model trends warranting careful study.
>
> ## "GNN" is more common (ad RC1)
>
> We use the term "GNN" since it is the most common term. GemNet and GemNet-OC still fall under the typical message-passing framework of GNNs, even if their update functions are more advanced. Whether the model operates on 2D or 3D data is immaterial to this framework. We do not think using a less common term would service the reader, especially since we are already very clear about the task and model in both the title and paper.
>
> ## Further baselines (ad RC2)
>
> As suggested, we have independently reimplemented and evaluated (relaxation-based) PaiNN [1] as an 8th baseline. Our reimplementation predicts forces directly from vectorial features via a gated equivariant block instead of using gradients of the energy output. This breaks energy conservation but is essential for good performance on OC20. This model obtains an average IS2RE MAE of 471meV. We have added this result to the paper.
>
> 3D-Graphormer [2] and SphereNet [3] only present direct IS2RE results in their papers, while all presented models are relaxation-based, i.e. they generate predictions based on relaxing the energy of an S2EF model. This is significantly more accurate but also more expensive than direct models. Adding direct models to the table would thus be rather misleading. That being said, we do mention 3D-Graphormer and its MAE (472meV vs. GemNet-OC's 355meV) in the main text. These results are substantially better than SphereNet's, which reports an average IS2RE MAE of 825meV.

---

### Decision · Action_Editors · 2022-09-21

**Recommendation:** Accept as is

**Comment:**

Although the work is of limited technical novelty, all the reviewers appreciate its solid empirical study (e.g., reduced training cost, sota results with significant improvements over baselines), which is very helpful to the community. Therefore, I recommend "accept as is".